# Network impact of a single-time-point microbial sample

**Shir Ezra, Amir Bashan** [ORCID] *

Physics Department, Bar-Ilan University, Ramat Gan, Israel

* amir.bashan@biu.ac.il

**Data Availability Statement:** The Python code used to calculate the network impact parameters, and a tutorial, are freely available at https://github.com/ezrashir/Network-Impact-of-a-single-sample.

## Abstract

The human microbiome plays a crucial role in determining our well-being and can significantly influence human health. The individualized nature of the microbiome may reveal host-specific information about the health state of the subject. In particular, the microbiome is an ecosystem shaped by a tangled network of species-species and host-species interactions. Thus, analysis of the ecological balance of microbial communities can provide insights into these underlying interrelations. However, traditional methods for network analysis require many samples, while in practice only a single-time-point microbial sample is available in clinical screening. Recently, a method for the analysis of a single-time-point sample, which evaluates its 'network impact' with respect to a reference cohort, has been applied to analyze microbial samples from women with Gestational Diabetes Mellitus. Here, we introduce different variations of the network impact approach and systematically study their performance using simulated 'samples' fabricated via the Generalized Lotka-Volttera model of ecological dynamics. We show that the network impact of a single sample captures the effect of the interactions between the species, and thus can be applied to anomaly detection of shuffled samples, which are 'normal' in terms of species abundance but 'abnormal' in terms of species-species interrelations. In addition, we demonstrate the use of the network impact in binary and multiclass classifications, where the reference cohorts have similar abundance profiles but different species-species interactions. Individualized analysis of the human microbiome has the potential to improve diagnosis and personalized treatments.

## Introduction

It has long been known that the human microbiome, i.e., the community of microbial species living on and within us, plays important roles in maintaining our health and well-being, including contribution to digestion, resistance to pathogens and vitamin production [1, 2]. Specifically, it has great potential to be used for personalized medicine [3–5]. First, each host bears distinct microbiome composition and species abundance [2, 6]. Thus, it might provide unique fingerprints of various conditions [7–9], and may assist with the prediction of patient-specific disease activity, manifestations, severity, and responsiveness to treatment [10].

**Funding:** This research was supported by the Israel Science Foundation [1258/21], the United States-Israel Binational Science Foundation (BSF), Jerusalem, Israel [2020255], and the German-Israeli Foundation for Scientific Research and Development [I-1523-500.15/2021] in the form of grants to AB.

**Competing interests:** The authors have declared that no competing interests exist.

Moreover, as opposed to genetic information, it can be actively manipulated for medical uses. Its alteration may affect human health and proper intestinal functions [11], drug administrated orally [12] and the physiological and pathological states of the host [13].

The abundances of the microbial species are determined by a network of ecological inter-species interactions of various mechanisms such as competition over limited resources, cross-feeding, and others [14, 15]. Analyzing this inter-species interrelations network can have important practical implications. First, studies show strong associations between certain abnormalities of the interaction networks and a variety of human diseases such as Crohn's disease, gestational diabetes, gastric carcinogenesis, etc. [16–20]. Thus, analysis of the microbial network, and its comparison to the known networks of healthy and diseased populations, may provide information about the health state of the patient. Second, the effects of active manipulation of the microbiome cannot be precisely predictable without an understanding of the changes it might inflict on the abundance of other species, which in turn may induce a dramatic effect on the entire microbial community.

Traditional approaches for analyzing the interaction network of individual patients rely on a prior inference of the network from a long time-series of samples, containing perturbations and responses, from each patient [21, 22]. However, in practice, multiple microbial samples are not collected during medical examinations as the process can be inconvenient and time-consuming. Furthermore, in most cases, the added value of collecting more than one sample from a patient might be limited, since the microbiome is mostly stable over time and changes mainly after strong perturbations such as antibiotic treatment, severe hunger or excessive alcohol consumption [23–25]. Consequently, since only one time point sample is available, the potential information concealed within microbial interactions is usually ignored.

Recently, a method for the analysis of a single-time-point sample that measures its 'network impact' with respect to a reference cohort of samples has been introduced and used to analyze microbial samples from women with Gestational Diabetes Mellitus [26]. The network impact measures how different are the species-species correlation networks of the reference cohort calculated with and without a single test sample. This approach is inspired by the LIONESS method developed for estimating the complete sample-specific regulatory networks in gene expression data [27], with the main difference that the network impact is a global measure that averages all inter-species interactions. The main point of the network impact assessment is that a test sample associated with similar inter-species interrelations as the cohort will have only a small effect on the network, while a test sample associated with different inter-species interrelations will have a large effect on the network.

Here, we extend the network impact method and methodically test its performance. We first introduce three different parameters for quantifying the network impact. Then, we systematically analyze the performance of the network impact method using simulated data generated by numerical simulations of the Generalized Lotke-Volterra (GLV) model, a mathematical model of ecological dynamics widely used in microbiome studies, e.g. in Ref. [28] (Fig 1a and Methodology). We simulate a cohort of 'microbial samples' as a set of alternative steady states of the same GLV model, using different initial conditions. We calculate the network impact of a single sample with respect to the cohort to test whether it was fabricated as another alternative steady state of the same GLV model, using only the abundance profile of the test sample. The network impact associated with an alternative steady state of the same GLV model as the reference cohort is expected to be considerably smaller compared with the network impact associated with a sample that was generated using a different GLV model.

In our study, we use two different classification setups (using terms borrowed from the machine learning jargon): "semi-supervised"—where all training samples belong to one type

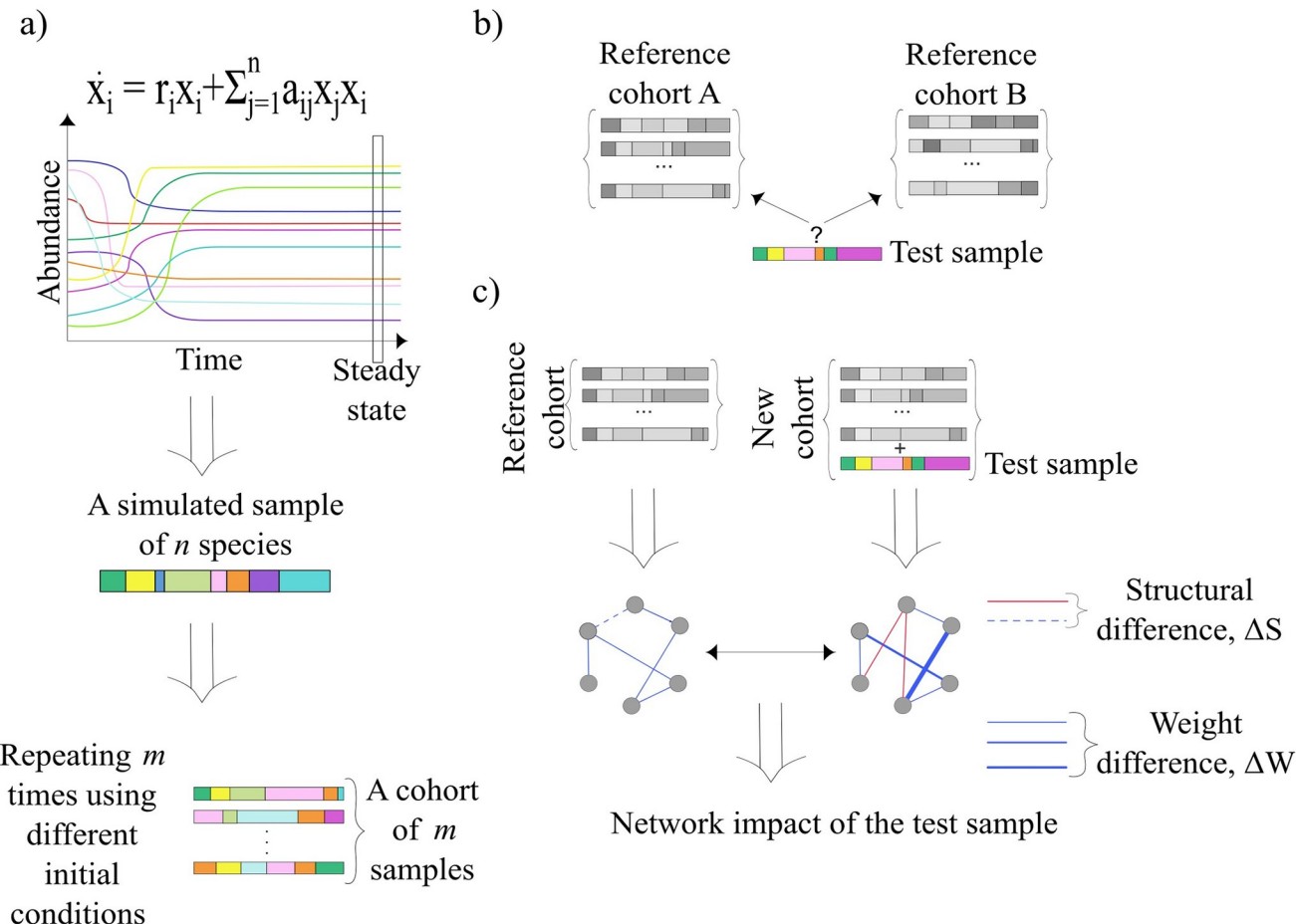

**Fig 1. Network impact approach for evaluating the underlying ecological dynamics of a single microbial sample.** a) The creation of a single sample via the time evolution of the GLV equations, defined as the species abundances after the system has reached a steady state. A cohort of samples is generated by repeating the previous calculation $m$ times with random initial conditions. b) Our goal is to classify a test sample, based only on its species abundance profile, into the cohort which was fabricated using the same GLV model. c) The network impact associated with the test sample measures how different are the species-species correlation networks of the reference cohort, calculated with and without the test sample. The three network impact parameters (see main text) consider the differences in the network structures and the weights of the links.

and a single test sample is tested according to its proximity to the same type (also referred to as "anomaly detection"); And "Supervised"—where two types of labeled training samples are available and the test sample is classified into either one. The semi-supervised setup may represent a scenario where a patient is compared to a healthy population, while the supervised setup represents a comparison of the patient's data to multiple populations with different health and disease conditions. To focus on the added value of the network impact as a measure of the species-species interrelations, we classify samples that share the same abundance distributions for each species independently but have different patterns of inter-species interrelations. We show that while these samples cannot be distinguished using the standard sample-sample dissimilarity measures, the three network impact parameters, which capture the effect of inter-species interactions, are able to serve as classifiers for these samples. In addition, we investigate the effect of the number of reference samples on the classification and compare the performances of the three different network impact parameters in the two different setups.

## Methodology

Generally, we calculate the network impact of a single test sample in order to classify it to its original cohort. In this section, we first describe the procedure of simulating samples using the GLV model, and the procedure of creating a cohort of shuffled samples. Then, we describe two approaches for evaluating a single sample: the calculation of the three network impact parameters and distance-based analysis.

### Numerical simulations of samples using the Generalized Lotka-Volterra (GLV) model

The GLV model describes the ecological dynamics of $n$ species using a set of ordinary differential equations. The change through time in the abundance of species $i$, $x_i$, is:

$$\frac{dx_i}{dt} = r_i x_i + \sum_{j=1}^{n} a_{ij} x_i x_j,$$ (1)

where $r_i$ represents the intrinsic growth rate of species $i$, and $a_{ij}$ represents the impact of species $j$ on species $i$. In our simulations, each off-diagonal element of the interaction matrix is drawn from a normal distribution $a_{ij} \in \mathcal{N}(0, \sigma^2)$ with probability $C$, and is zero otherwise, and the diagonal elements are set to be $a_{ii} = -1$. The intrinsic growth rate of species $i$ is drawn from a uniform distribution $r_i \in \mathcal{U}(0, 1)$. We consider a microbial 'sample' as a steady state of a GLV model (Fig 1a). For a particular GLV model, a cohort of $m$ samples is simulated as a set of alternative steady states of Eq (1), integrated using random initial species assemblages and abundances. Specifically, the initial abundance of each species is drawn from a uniform distribution $x_i(t = 0) \in \mathcal{U}(0, 1)$ with probability $p = 0.8$, and is zero otherwise. We represent the cohort of samples as an $m \times n$ matrix $X$, where $X_{ki}$ represents the steady-state abundance of species $i$ in sample $k$ (Fig 1a). Throughout our simulations, we use $n = 100$ and specify the value of $m$ in each experiment. Finally, we normalize the samples such that the abundances of all species in each sample sum to one, i.e., in terms of computational notations, $X_{ki} := X_{ki}/\Sigma_i X_{ki}$, where $:=$ is the assignment expression operator.

### Shuffled data

For the semi-supervised analysis, we fabricate test 'samples' that contain no inter-species interrelations by shuffling the species abundances of a cohort of GLV samples for each species independently. This method preserves the species assemblages and abundance distribution of the original samples but eliminates any species-species correlations. We generate a new matrix of shuffled samples $S$ of size $m \times n$ based on the $m \times n$ matrix of the original samples $X$, by applying the Fisher-Yates shuffling algorithm on each column of $X$, which represents the abundances of a particular species in the $m$ different samples (See Fig 2). The shuffling of each

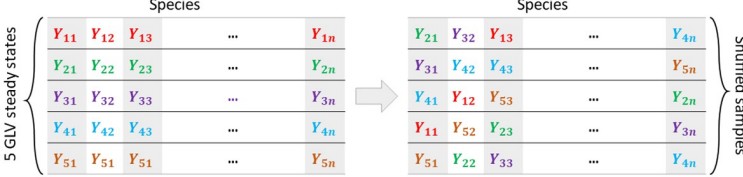

**Fig 2. Schematic demonstration of the shuffling procedure.** The right cohort of shuffled samples was fabricated by shuffling the cohort of GLV steady states on the left.

column is performed as follows: for $k$ from $m$ down to 1:

$$S_{ki} = \begin{cases} 0 & X_{ki} = 0 \\ X_{kv} & otherwise \end{cases} \tag{2}$$

where $v$ is a random integer $1 \leq v \leq k$. Finally, each shuffled sample in $S$ is normalized to one such that

$$\sum_{j=1}^{n} S_{kj} = 1, \tag{3}$$

for sample $k$ ($k = 1, \ldots, m$).

## Evaluating a single microbial sample

**Network impact of a single sample.**   The network impact of a single test sample is calculated by comparing two networks: one created from a reference cohort of $m$ samples (referred to as "Network $N^{(m)}$") and the other from a new cohort of $m + 1$ samples that includes both the reference samples and the test sample (referred to as "Network $N^{(m+1)}$") ([Fig 1c]). The network impact assesses how well the proportions of species abundances in the test sample align with the pattern of inter-species interrelations observed in the reference cohort. We quantify the network impact in terms of dissimilarity between the two networks. A low value of network impact suggests that the inter-species interactions in the local community associated with the test sample are similar to those in the reference samples. On the other hand, a high value of network impact suggests that the ecological balance of the test sample is associated with different underlying interactions than those of the reference samples.

To construct a network $N$ for a given cohort, we first calculate the $n \times n$ matrix of the Pearson correlation coefficients and their corresponding p-values, for each pair of species (See [Methods]). For the calculation of each matrix element $\rho_{ij}$ ($i, j = 1, \ldots, n$), we use only the subset of samples where both species ($i$ and $j$) have non-zero abundances, i.e., a sample $k$ is included in the calculation of $\rho_{ij}$ only if both $X_{k,i} > 0$ and $X_{k,j} > 0$. This is done to focus on the abundance variations that are directly and indirectly resulted by the inter-species interactions, while in our simulations the absence of a species in a given sample is randomly assigned in the initial conditions. Finally, the network $N$ is defined with the link's weights being $N_{ij} = \rho_{ij}$ if the corresponding p-value is $< 10^{-3}$, and $N_{ij} = 0$ otherwise.

Three parameters are used to quantify the distinctions between network $N^{(m)}$, created from the reference cohort, and network $N^{(m+1)}$, created from the reference cohort and the test sample. Initially, we evaluate the extent to which the structure of network $N^{(m+1)}$ differs from that of network $N^{(m)}$. The *structural difference*, $\Delta S$, is defined as the Jaccard dissimilarity between the unweighted networks

$$\Delta S = 1 - \frac{|L|}{|N^{(m)} \cup N^{(m+1)}|} \tag{4}$$

where $L \equiv N^{(m)} \cap N^{(m+1)}$ is the set of shared links in $N^{(m)}$ and $N^{(m+1)}$, i.e., the set of $ij$ for which $N_{ij}^{(m)} \neq 0$ and $N_{ij}^{(m+1)} \neq 0$. If the unweighted structure of $N^{(m)}$ is identical to the one of $N^{(m+1)}$, $\Delta S = 0$.

Second, we focus on the shared links (the set $L$) and assess the degree of variation in their weights. The *differential weight difference*, $\Delta W$, is the average difference between the weights of

the shared links

$$\Delta W \equiv \frac{1}{|L|} \sum_{i,j \in L} N_{ij}^{(m)} - N_{ij}^{(m+1)}. \tag{5}$$

Thus, $\Delta W = 0$ means that the weights of the shared links remain unchanged in the two networks.

Lastly, $\theta$ is defined as the proportion of network links whose weights decreased after incorporating the single test sample into the reference cohort. The last network impact parameter $\theta$ is complimentary to $\Delta W$ in the sense that $\Delta W$ represents the magnitude of change in the networks while $\theta$ determines the direction of change, that is, whether the additional sample is associated with a general increase or decrease in the inter-species interrelations.

$$\theta = \frac{|H|}{|G|}, \tag{6}$$

Where $G$ represents the subset of shared links (subset of $L$) whose weights differ between network $N^{(m)}$ and network $N^{(m+1)}$ ($G = \{N_{ij}^{(m)} | N_{ij}^{(m)} \in L; N_{ij}^{(m)} \neq N_{ij}^{(m+1)}\}$), and $H$ is a subset of $G$ containing all the links with decreased weights in $N^{(m+1)}$ compared to $N^{(m)}$ ($H = \{N_{ij}^{(m)} | N_{ij}^{(m)} \in G; N_{ij}^{(m+1)} < N_{ij}^{(m)}\}$).

In this case, a test sample associated with similar inter-species interactions as the reference cohort will yield only random fluctuations in the weights of the networks' links, and thus a value of $\theta \approx 0.5$. In contrast, a test sample associated with very different inter-species interactions from the reference cohort is expected to lead to a larger fraction of decreased correlations in $N^{(m+1)}$, yielding $\theta > 0.5$.

**Distance-based analysis.** A more traditional way of testing the relations between a test sample and a reference cohort is by analyzing its average sample-sample distance from the cohort's samples. The average distance of the test sample should be compared with the distribution of the sample-sample distances between the cohort's samples themselves. A significantly larger average distance of the test sample, represents an anomaly of the test sample with respect to the reference cohort, e.g., it may indicate that the sample has been shaped by different ecological rules. Similarly, the classification of a single test sample to one out of several reference cohorts can be done by choosing the closest cohort, i.e., the cohort with the minimal associated average distance. From the various available dissimilarity measures, we choose to use the root Jensen Shannon Divergence (rJSD), because it is a distance metric that satisfies non-negativity, identity, symmetry and triangle inequality.

Given two samples, **p** and **q**, the rJSD is defined as:

$$rJSD(\mathbf{p}, \mathbf{q}) = \left[ \frac{D_{KL}(\mathbf{p}\|\mathbf{m}) + D_{KL}(\mathbf{q}\|\mathbf{m})}{2} \right]^{1/2} \tag{7}$$

Where $\mathbf{m} = \frac{\mathbf{p}+\mathbf{q}}{2}$ and $D_{KL} = \Sigma_i p_i log\left(\frac{p_i}{q_i}\right)$ is Kullback-Leibler divergence between $p$ and $q$.

## Results

### Semi-supervised classification of a single sample

In this section, we aim to determine if a single test sample is fabricated as an alternative steady state of a particular GLV model or as a shuffled profile. The shuffled profiles may represent microbial communities associated with a neutral ecological model where species differ in their characteristic self-dynamics (growth rates and species-environment interactions) but with no

significant inter-species interaction. Since the shuffling procedure preserves the statistics of each species independently but removes the effect of the inter-species interrelations, distinguishing between GLV steady states and shuffled samples indicates that the effect of inter-species interrelations can be detected from single-time-point data. We begin by simulating a reference cohort of $m$ = 500 steady states ('samples'), using a GLV model defined by a particular set of growth rates $r_i$ and inter-species interactions $a_{ij}$ (with the values of $\sigma$ = 0.2 and $C$ = 0.1). We then introduce a single test sample, which can be either generated as an alternative steady-state of the same GLV model or taken from a shuffled cohort (see Methods). We prepare 2, 000 test samples, 1, 000 GLV steady states and 1, 000 shuffled profiles.

We test whether a single sample can be classified based on the sample-sample dissimilarity between it and the reference cohort. We measure the average sample-sample dissimilarity (rJSD, see 'Distance-based analyses' in Methods) between each test sample and the samples of the reference cohort. Fig 3a shows that the two distributions of sample-sample dissimilarity values calculated for the GLV steady states and for the shuffled profiles have the same mean values ($\approx$ 0.4) and largely overlap (p-value > 0.63, using t-test). This demonstrates the limitation of a distance-based measure to detect information embedded in the correlations between the species.

Fig 3b shows the distributions of network impact parameter $\theta$ values for the two types of test samples. The network impact values of the GLV steady states are distributed around $\theta$ = 0.5. This is expected as they are generated using the same interactions. In contrast, the average network impact value of the shuffled profiles is higher than 0.5, as the inter-species correlations in each shuffled sample are very different from the cohort samples. In addition, the two distributions of the GLV steady states and the shuffled profiles are significantly different, with a p-value $< 10^{-32}$ using a t-test. This indicates a clear distinction between the two types of test samples and may allow us to distinguish a single shuffled sample from a cohort of GLV steady states using the network impact parameter $\theta$.

To quantify the separation between the different distributions, we plotted a Receiver Operating Characteristic (ROC) curve for both the species-species dissimilarity and the network impact parameter $\theta$. Fig 3c illustrates that the classification of a test sample using the species-species dissimilarity will yield similar results to that of a random classifier, as indicated by an

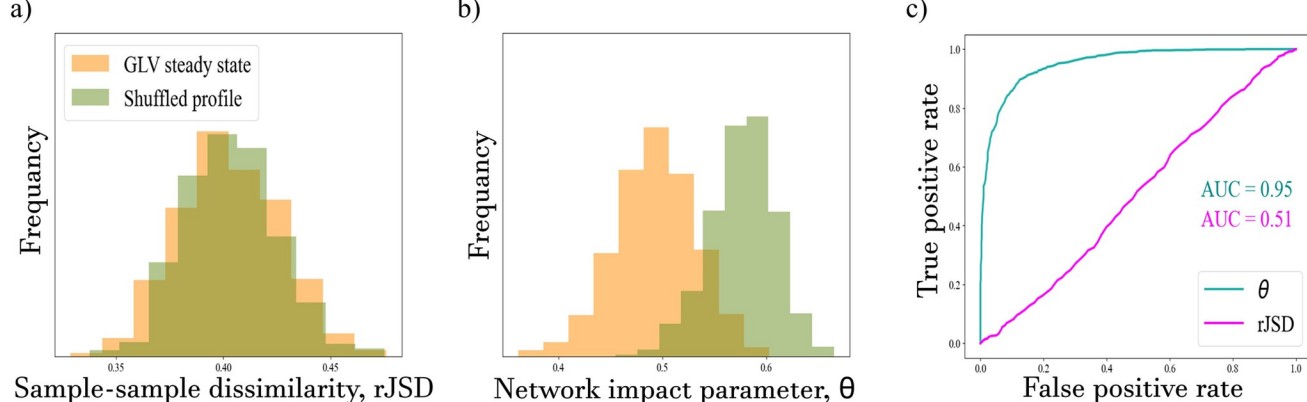

**Fig 3. Network impact distinguishes real from shuffled samples in a semi-supervised classification setup.** a) Distributions of the sample-sample dissimilarity (rJSD) values calculated for all single test samples, GLV steady states and shuffled profiles, with respect to the reference cohort. b) Distributions of $\theta$ values of GLV steady states and shuffled profiles. The separation between the two distributions suggests that the $\theta$ value can be used to classify a single sample. c) A Receiver Operating Characteristic (ROC) curve for the classification using the sample-sample dissimilarity measure (pink) and the network impact parameter $\theta$ (cyan).

Area Under the Curve (AUC) of about 0.5. In marked contrast, the test sample can be effectively classified using the network impact parameter $\theta$, as indicated by an AUC of about 0.95.

This result brings to light a practical question regarding the size of the reference cohort, $m$, required for a successful classification using the network impact parameters. On the one hand, with more samples in the reference cohort, the reference network contains more available information that is used to calculate a more accurate reference network. On the other hand, the larger the number of samples in the reference cohort, the lower the relative weight of a single test sample. To address this question, we systematically calculated the network impact parameters for different reference cohort sizes. For each cohort size, $m$, we calculated the network impact values for 100 shuffled profiles and 100 GLV steady states, as shown in Fig 4.

Our analysis of the network impact parameters for different reference cohort sizes, as shown in Fig 4, reveals several key findings. Firstly, Fig 4a–4c demonstrates that the median values of all network impact parameters for the shuffled profiles are higher than those of the GLV steady states for all values of $m$. Furthermore, it is clear that when the reference cohort size is $m = 25$, the distributions overlap (for each network impact parameter), as indicated by relatively low AUC values. This suggests that an effective classification using network-impact analysis cannot be achieved with a cohort of only 25 samples.

Fig 4a shows the values of the structural difference $\Delta S$, and Fig 4b shows the differential weight difference $\Delta W$ for different reference cohort sizes. When considering both $\Delta S$ and $\Delta W$, it is evident that in our simulations the distributions for the cases of $m = 50$ and $m = 100$ have the highest AUC values. This suggests that there is an optimal range of $m$ values for maximizing the separability between real and shuffled profiles using $\Delta S$ and $\Delta W$, where the analysis of a small reference cohort can result in a noisy network, and a large reference cohort may mask

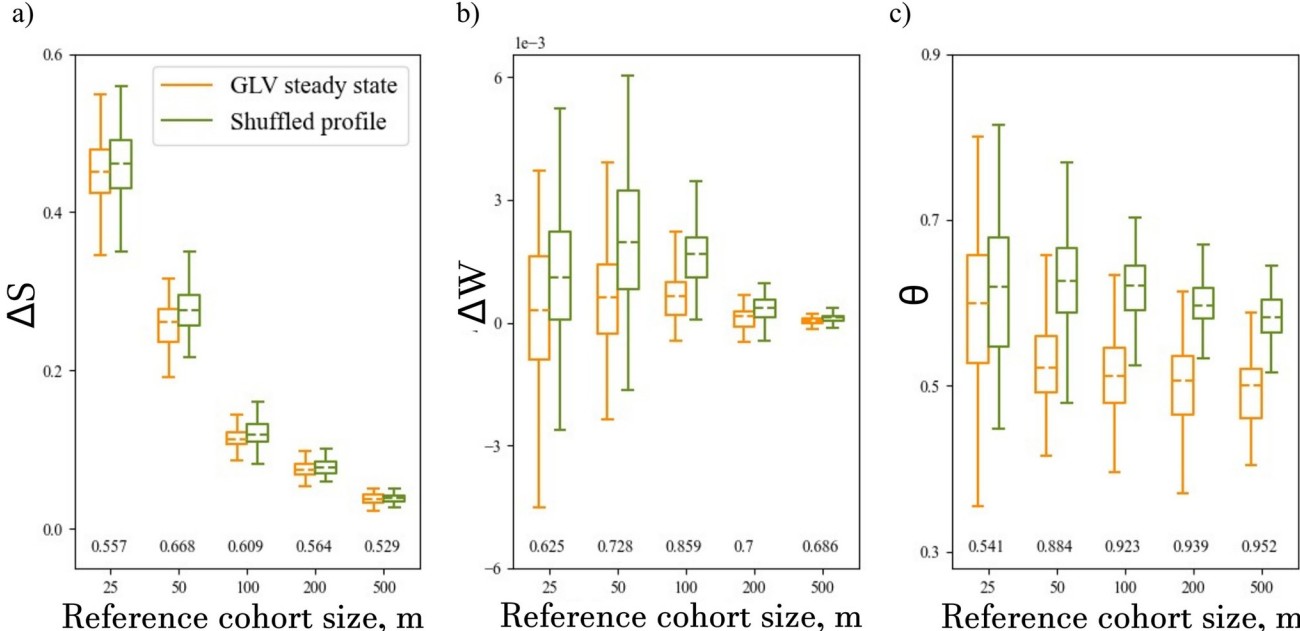

**Fig 4. The optimal size of the reference cohort for calculating the network impact of a single sample.** Comparison of the $\theta$ values distributions of the GLV samples (green) and the shuffled profiles (orange). Each box extends from the first quartile to the third quartile of the data, with a line at the median. The whiskers extend from the box by 1.5 the inter-quartile range. Each pair of distributions were calculated with respect to a reference cohort with $m$ samples. The AUC value for each pair of distributions is reported beneath. a) For $\Delta S$, the AUC values are the highest for $m = 50$ and $m = 100$. b) For $\Delta W$, the AUC values are the highest for $m = 50$ and $m = 100$. c) For $\theta$, the AUC values increase for larger $m$, and for $m \geq 50$ they are larger compared with those of the other network impact parameters.

the relative network impact of a single sample. In cases where many reference samples are available, applying the network impact parameters $\Delta S$ or $\Delta W$ directly on the large reference cohort may be ineffective. Rather, they can be calculated over sub-groups of the reference samples, whose size should be chosen according to the optimal range of $m$.

In contrast to $\Delta S$ and $\Delta W$, the AUC values for $\theta$ do not show an optimal range of reference size but rather monotonically increase with $m$ (Fig 4c). This difference stems from the fact that $\Delta S$ and $\Delta W$ measure the magnitude of the effect of the test sample on the reference network, while $\theta$ measures the direction of this change (in terms of increased versus decreased correlations), regardless of its magnitude. In addition to the independence of $\theta$ from $m$, it has considerably higher AUC values than the other two network impact parameters, suggesting its supremacy as a classifier in this setup.

## Supervised classification of a single sample

In this setup, we classify a single test sample into one of several reference cohorts generated using different GLV models. Each reference cohort consists of $m = 50$ simulated alternative steady states ('samples'). All GLV models are created using the same set of $r_i$ values, such that the characteristic abundance of each species is preserved across the different models. The alternative case, where the individual species have different growth rates, and consequently, differential abundance profiles, can be easily classified using distance-based analysis of the abundance profiles, without the need to assess the species-species interrelations. However, the interaction matrices ($a_{i,j}$ for $i \neq j$) are unique to each model, chosen using the parameters $\sigma = 0.01$, $C = 0.8$ (see Methodology section). Here, we choose a large value of $C$ and weak interaction strength (low value of $\sigma$) to reduce the accumulated effect of the inter-species interactions on the characteristic abundance of the individual species. In other words, when all species have many weak interactions, we expect that, on average, the contributions of the positive and negative interactions on the species abundance will cancel out. Each test sample is fabricated as an alternative steady state of one of the models that are used for the reference cohorts. The classification task is to identify its associated model based on its abundance profile alone.

We begin with a binary classification task, where two GLV models ('model A' and 'model B') are used to simulate two reference cohorts, as well as 200 test samples (100 test samples from each model). Fig 5a shows the average sample-sample dissimilarity (rJSD) calculated between each test sample and each of the two reference cohorts. The distributions of dissimilarity values for test samples of model A (orange points) with respect to each model are effectively the same (the values are evenly distributed around the equality line). The same results are obtained for test samples of model B (gray points). In this case, the analysis of the sample-sample dissimilarity alone cannot be used to identify the original reference cohort of a test sample. This is because the sample-sample dissimilarity measure focuses on the species abundances, which are similar in both models, but does not consider their model-specific inter-species interrelations.

Next, we calculated the three network impact parameters, $\Delta S$, $\Delta W$, and $\theta$, for each test sample with respect to each reference cohort. Fig 5b and 5c demonstrates a clear separation between the clusters of test samples generated from the different GLV models when considering the network impact parameters $\Delta S$ and $\Delta W$. Test samples simulated using model A have a relatively small network impact with respect to reference cohort A and a high network impact with respect to reference cohort B. This results in distinct clusters that enable us to effectively classify an unknown test sample to its original GLV model.

Unlike the other two network impact parameters, $\Delta S$ and $\Delta W$, the network impact parameter $\theta$ displays an area of overlap (Fig 5d). Test samples of model A have typically high $\theta$ values

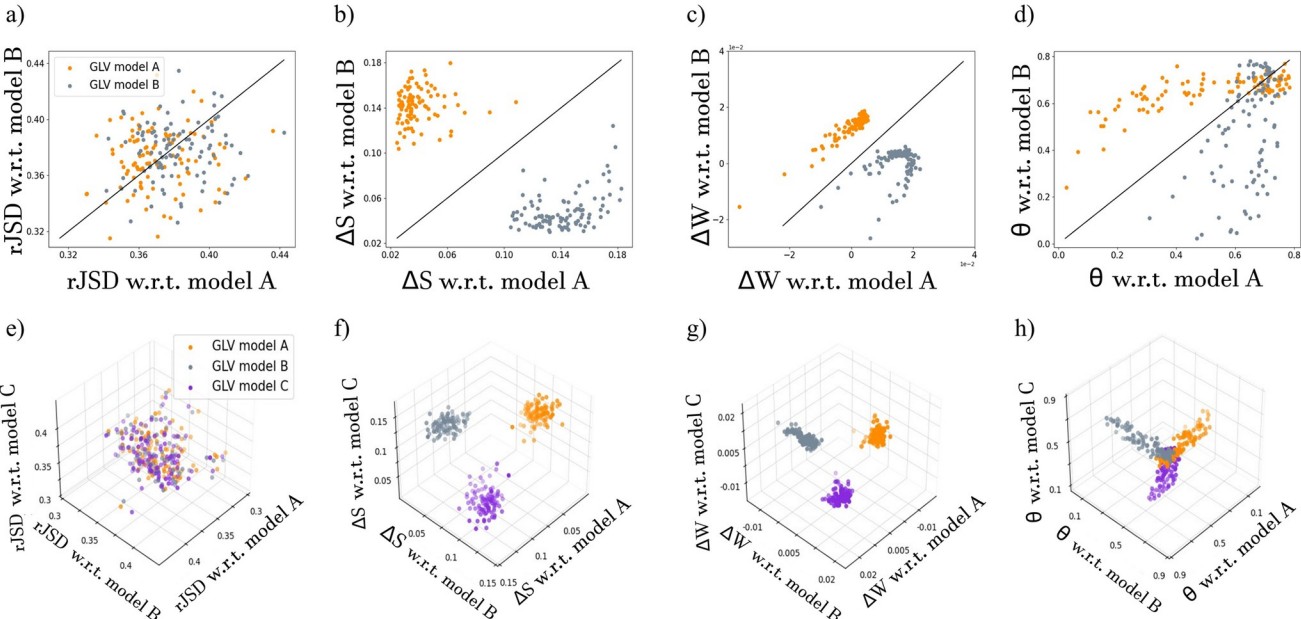

**Fig 5. Distance-based versus network impact parameters for binary and multiclass supervised classification.** a) Each point represents the average dissimilarity (rJSD) calculated for a single test sample with respect to each of the reference cohorts ('model A' and 'model B'). Orange and gray points represent test samples from models A and B, respectively. The black line is the equality line, such that points above the equality line represent samples that are more similar to model A than to model B, and vice versa. b-d) Each point represents the network impact parameters ($\Delta S$, $\Delta W$ and $\theta$) calculated for a single test sample with respect to each of the reference cohorts ('model A' and 'model B'). Points above the equality line represent samples with lower network impact with respect to model A than with respect to model B, and vice versa. e-h) Similar to panels a-d for the multiclass classification, where the reference cohorts and test samples are generated using three different GLV models. The results for all panels were obtained using the same data, i.e., reference cohorts and test samples.

with respect to model B, while the $\theta$ values with respect to model A are distributed around 0.5, as expected, but with a large variance. Similar results are obtained for test samples of model B. This result suggests that $\theta$ is less effective compared with $\Delta S$ and $\Delta W$ in classifying single test samples in the supervised setup.

Finally, we demonstrate the performances of the network impact parameters in the multiclass classification setup, where the reference cohorts and the test samples are simulated using three different GLV models. Similarly to the binary case, the results demonstrate that the test samples cannot be distinguished using the sample-sample dissimilarity (Fig 5e) but can be effectively classified using the network impact parameters $\Delta S$ and $\Delta W$ (Fig 5f and 5g), and less effectively using $\theta$ (Fig 5h).

## Discussion

The network impact parameters perform differently in the different setups. In the semi-supervised setup, GLV steady states are best distinguished from shuffled profiles using the $\theta$ parameter. In contrast, in the supervised analyses, the $\Delta S$ parameter performs best. Both network impact parameters are designed to detect abnormalities in the pattern of species-species interactions compared with the reference cohort. Yet, the difference between their performances can be explained as follows. The $\Delta S$ parameter is designed to capture large abnormalities, which are associated with structural differences in the correlation networks. However, it is less sensitive to more minor abnormalities as it disregards differences in the weights of the

networks' links. In contrast, the $\theta$ parameter focuses on the changes in the link's weights, which are associated with more minor abnormalities. Furthermore, unlike the $\Delta W$ which measure the 'magnitude' of the impact, the $\theta$ measure only its 'direction' and thus has the advantage of high performance in cases of large reference cohort sizes. Therefore, in our supervised classification setup, where the samples are generated with two different GLV models, the abnormality of a single sample with respect to the other model is typically large and best captured by the $\Delta S$ parameter. The case of distinguishing real GLV steady states from the shuffled profile is associated with more minor abnormalities and thus is best classified using the $\theta$ parameter.

Previous studies have employed supervised analysis of the microbiome using machine learning (ML) techniques [29]. A main limitation of such techniques is that they typically require a large amount of training data. In contrast, the network impact can be applied using a relatively small number of reference samples. Another qualitative difference is that, unlike ML techniques which usually do not incorporate any prior knowledge, the network impact approach can be applied with the knowledge that the network impact of a single sample with respect to its associated cohort is expected to be very low. As a result, the network impact can be easily used for semi-supervised anomaly detection tasks, where only a single reference cohort is available.

## Conclusions

In this paper, we study how the concept of network-impact can be used for the analysis of single-time-point microbial samples, revealing information that may be inaccessible using a distance-based approach, in order to facilitate microbiome-based personalized medicine [3, 5]. We proposed three different network impact parameters and tested their performances as classifiers in semi-supervised anomaly detection and supervised classification setups, using GLV simulations. These setups represent different scenarios in the clinical diagnosis process based on the analysis of a single sample. The semi-supervised classification setup of single-time-point samples represents a situation in which a microbial sample from a patient is tested against only a healthy population but not against a particular disease. The supervised classification setup represents a situation where a single test sample is compared to reference cohorts of distinct health conditions.

In our experiments, we intentionally simulated samples that cannot be easily classified based on their similarity to the reference cohorts, in order to focus on the added value of the network impact. In practice, a recommended way to study an unknown single sample is to first examine it via traditional statistical tests, such as measuring the similarity between the test sample and the reference cohort. If the similarity-based methods cannot, or only partially, classify the data, the network impact may offer additional perspectives for the analysis. Another optional direction is to incorporate information on the relative relatedness of community members by incorporating phylogenetic distances between observed organisms in the computation using the Unifrac distance [30], or to leverage the Genomic Content Network to measure the functional redundancy of a single sample [31].

We introduced our network impact analysis within the context of the human microbiome. Beyond the human microbiome, this method can be directly applied to various other microbial environments. For instance, it can be applied to soil, ocean, lakes, phyllosphere/rhizosphere, bioreactors, and fermenters particularly when dealing with scenarios where only a single microbial sample is accessible or when conducting longitudinal measurements presents practical challenges. Thus, our methodology offers an adaptable tool for assessing microbial interactions across different ecosystems.

## Acknowledgments

The authors thank Yogev Yonatan, Guy Amit, Yakir Perez and Liad Shamir for their helpful comments and discussions.

## Author Contributions

**Conceptualization:** Amir Bashan.

**Formal analysis:** Shir Ezra.

**Funding acquisition:** Amir Bashan.

**Investigation:** Shir Ezra.

**Methodology:** Amir Bashan.

**Project administration:** Amir Bashan.

**Software:** Shir Ezra.

**Supervision:** Amir Bashan.

**Visualization:** Shir Ezra.

**Writing – original draft:** Shir Ezra, Amir Bashan.

**Writing – review & editing:** Shir Ezra, Amir Bashan.

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
