## [Decision Letter · Decision Letter 0]

11 Mar 2024

PONE-D-23-16744Network impact of a single-time-point microbial samplePLOS ONE

Dear Dr. Bashan,

Thank you for submitting your manuscript to PLOS ONE. After careful consideration, we feel that it has merit but does not fully meet PLOS ONE’s publication criteria as it currently stands. Therefore, we invite you to submit a revised version of the manuscript that addresses the points raised during the review process.

I am writing to inform you that I have recently been appointed as the academic editor for your manuscript submission. Upon reviewing the history of the submission process, I have observed that there have been challenges in securing reviewers for your manuscript. I would like to extend my sincere apologies for any inconvenience this may have caused you.

I am pleased to share that, from the reviewers who have agreed to evaluate your work, both have expressed a positive outlook towards your manuscript. We concur that with minor modifications, your article could be well-positioned for acceptance. 

We look forward to receiving your revised manuscript.

Kind regards,

Luis D. Alcaraz, Ph.D.

Academic Editor

PLOS ONE

Journal Requirements:

"The authors thank Yogev Yonatan, Guy Amit, Yakir Perez and Liad Shamir for their helpful comments and discussions. This research was supported by the Israel Science Foundation (grant No. 1258/21) and the German-Israeli Foundation for Scientific Research and Development (grant No. I-1523-500.15/2021)."

"AB thanks the Israel Science Foundation (grant No. 1258/21) and the German-Israeli Foundation for Scientific Research and Development (grant No. I-1523-500.15/2021) for supporting this research.

Reviewers' comments:

Reviewer's Responses to Questions

**Comments to the Author**

1. Is the manuscript technically sound, and do the data support the conclusions?

Reviewer #1: Yes

Reviewer #2: Yes

2. Has the statistical analysis been performed appropriately and rigorously? 

Reviewer #1: Yes

Reviewer #2: Yes

3. Have the authors made all data underlying the findings in their manuscript fully available?

Reviewer #1: Yes

Reviewer #2: Yes

4. Is the manuscript presented in an intelligible fashion and written in standard English?

Reviewer #1: Yes

Reviewer #2: Yes

5. Review Comments to the Author

Reviewer #1: The manuscript “Network impact of a single-time-point microbial sample” reports new methods of analysis of microbial community composition. Prior work has analyzed how microbial community composition changes over time. Correlations within such networks reveal the interactions between different microbial species and the impact of these interactions on changes in species abundance over time. However, in many relevant contexts, it is difficult to take time-course measurements of community composition. The reported methods attempt to solve this problem by comparing the community composition at one time point with the characteristic composition of similar communities from prior measurements. The hope is that such comparisons might identify when a community is abnormal, in that the pattern of species abundances are not expected given the underlying interaction network in the comparison samples. Here data used for these comparisons are generated using generalized Lotka-Volterra models. Data generated from the same matrix of growth rate and interaction parameters are combined to create the reference cohort. This standard dataset is then rearranged to approximate individual samples that should disagree with the model used to generate the standard cohort. Several different analysis schemes are compared, with distance-based analysis representing a standard approach.

Overall the findings of the paper are interesting. The authors clearly demonstrate that the proposed analytical methods do reveal samples that differ from the reference cohort. Minor changes should be made to improve clarity of the manuscript, as described below.

1. The notation is the paper is at times confusing, particularly the use of i and j to mean multiple things. For example, in equation 1 i and j are different species in the same community, whereas later around line 99 j is a species and i is a sample number. Later on line 131 k indicates the sample number. Particularly in the paragraph starting on line 129 it is not clear for calculations of N_ij and p_ij if species abundances are compared only within a community or between different community samples. More explanation and consistent notation would help.

2. For the semi-supervised case C = 0.1, which means 90% of species combinations are not interacting. However for the supervised case C = 0.8, which means only 20% of species combinations are not interacting. It is unclear if this seemingly large change greatly impacts the ability of the metrics proposed to identify abnormal population ratios, but a note probably should be added explaining the need for the increase in C when switching to the supervised case.

3. On line 246 it is stated that “a large reference cohort may mask the relative network impact of a single sample”, but that seems like a failure of the analytical approach not a general problem of having too much initial data. Having more cohort data should improve the ability to identify abnormal sample, not make it more difficult. Is there a version of this analyze the corrects for the size of the reference cohort?

4. It seems like there would be some value to testing the ability of these metrics to identify samples generated from a GLV model using a different growth rate and interaction matrix, as opposed to only shuffling data generated from the same model parameters. Likely in real contexts, changes in external conditions or species genetic variation may modify these matrices, which leads to changes in the patterns of species abundance for some samples/patients.

Minor comments:

Page 1: “vitamins produce” should probably be vitamin production.

On pages 3 and 5 the phrase “sum to a unit” is used. Sum to one is more familiar to me, although sum to a unit may be acceptable.

On line 101 it states X_ij := X_ij / sum X_ij. Isn’t that a recursive definition of X_ij?

On line 112, I believe it should be: k is a random integer 1 <= k <= m not 1 <= k <= i.

Line 223: “This result arises a practical question” should be “This result suggests a practical question” or “This result brings to light a practical question”.

Reviewer #2: Reviewed the manuscript titled "Network impact of a single-time-point microbial sample". The authors present a method to estimate the divergence between a single sample and a cohort of samples that may represent different conditions, then compare this procedure with more traditional distance-based measures, and finally show two different approaches to use their method. The methods are technically sound and the data support the conclusions. The statistical analysis is appropriate. Both the data and code used in this manuscript are readily available in the indicated repository. The language used in the manuscript is clear and the text looks well prepared.

Recommended minor revisions to improve the manuscript for readability and attend a suggestion on the presentation of the story: The authors give much weight to the human microbiome and the application of their method for personalized medicine. Maybe presenting the problem being addressed by this method in a wider context could attract researchers from other fields like biotechnology (population dynamics within bioreactors), agriculture (state of rhizospheric communities), and microbial ecology in general (impact of climate change in a specific system). Besides, the applications on human microbiome research are barely mentioned in the conclusions. It is understandable if the authors want to keep the focus in the human microbiome, if that is the case they should further discuss the implications for this field.

More specific suggestions are given below:

- Remove the expression of the GLV model from the introduction and refer to methods or Figure 1

- Maybe add a visual explanation for the three parameters in Figure 1 or 2

- Line 211, replace "if"

- Text in all the plots is too small, please enlarge it

- Text in Figure 3 looks narrow, please enlarge/change the typography

- Rename the current "Conclusion" section to "Discussion" and add a concise "Conclusion" section with the main take-home messages

6. PLOS authors have the option to publish the peer review history of their article (what does this mean?). If published, this will include your full peer review and any attached files.

Reviewer #1: No

Reviewer #2: No

---

## [Author Response · Author response to Decision Letter 0]

19 Mar 2024

Our point-by-point responses are in the Response to Reviewers file

---

## [Editor Report · Decision Letter 1]

20 Mar 2024

Network impact of a single-time-point microbial sample

PONE-D-23-16744R1

Dear Dr. Bashan,

We’re pleased to inform you that your manuscript has been judged scientifically suitable for publication and will be formally accepted for publication once it meets all outstanding technical requirements.

Kind regards,

Luis D. Alcaraz, Ph.D.

Academic Editor

PLOS ONE

---

## [Editor Report · Acceptance letter]

17 May 2024

PONE-D-23-16744R1 

PLOS ONE

Dear Dr. Bashan, 

I'm pleased to inform you that your manuscript has been deemed suitable for publication in PLOS ONE. Congratulations! Your manuscript is now being handed over to our production team.

Kind regards, 

on behalf of

Prof. Luis D. Alcaraz 

Academic Editor

PLOS ONE